



# Ambiguous agricultural drought: characterising soil moisture and vegetation droughts in Europe from earth observation

Theresa C. van Hateren[1,2], Marco Chini[1], Patrick Matgen[1], and Adriaan J. Teuling[2]

[1]Luxembourg Institute of Science and Technology, Department of Environmental Research and Innovation, Esch-sur-Alzette, Grand-Duchy of Luxembourg
[2]Wageningen University and Research, Department of Hydrology and Quantitative Water Management, Wageningen, The Netherlands

**Correspondence:** T. C. van Hateren (tessa.vanhateren@list.lu)

**Abstract.** Climate change will likely lead to more regular and more severe drought events in the near future, with large impacts on agriculture, especially during long-lasting precipitation deficits or heat waves. This study focuses on agricultural droughts, which are generally defined as soil moisture deficits so severe, that vegetation is negatively impacted. However, during short soil moisture drought events, vegetation is not always negatively affected, and sometimes even thrives under these conditions. Because of this duality in agricultural drought impacts, the use of the term *agricultural droughts* is ambiguous. Here we show that, in major European droughts over the past two decades, clear asynchronies and discrepancies occur between soil moisture and vegetation anomalies. A clear delay is visible between the onset of soil moisture drought and vegetation drought, and correlation between the two types of drought generally peaks at the end of the growing season. This behaviour seems to be different in droughts at lower latitudes, where correlations peak earlier in the season, likely due to water limited conditions occurring much earlier there. Moreover, results indicate that in some cases, vegetation can show a positive anomaly, even when soil moisture anomalies are negative. As a result, the use of the term agricultural drought could lead to misclassification of drought events and false drought alarms depending on whether vegetation or soil moisture is used to quantify the drought. We argue that it is necessary to make a distinction between soil moisture drought and anomalies in vegetation.

## 1 Introduction

Due to climate change and enhanced land-atmosphere feedbacks, the impact of droughts will likely become more severe over the coming decades (Rasmijn et al., 2018; Samaniego et al., 2018; Teuling, 2018). Droughts are generally considered to be induced by a precipitation deficit relative to normal conditions, which, when persisting over longer time periods, results in insufficient water supply to meet demands of both human activities and the environment (Hayes et al., 2011). As a result, impacts of droughts can range from decreased crop yield and damage to ecosystems, to land subsidence, insufficient drinking water, and disruption of transport. To monitor and quantify drought across the terrestrial part of the hydrological cycle, numerous drought indices have been developed over the past decades. These can be divided into indices for the three main drought types. Meteorological droughts are defined as a prolonged period with below-normal precipitation. These droughts are typically quantified with the Standardized Precipitation Index (SPI, McKee et al., 1993), reflecting the current dogma





that droughts are measured relative to the mean climate as well as the climate variability at that location. Meteorological
droughts can propagate into hydrological droughts (Kumar et al., 2016), which entail below-normal (ground)water levels or
river discharge (Seneviratne et al., 2012), and are generally evaluated using e.g. reservoir levels, Standardized Runoff Index
or the Streamflow Drought Index (Shukla and Wood, 2008; Hayes et al., 2011). Lastly, agricultural droughts are defined as
a soil moisture deficit severe enough to hamper vegetation growth (Wilhite and Glantz, 1985). Due to their direct relation to
food production (through crop yield) and water management (through irrigation), agricultural drought is often the key focus of
drought monitoring and forecasting.

In line with their definition, agricultural droughts have traditionally been quantified based on soil moisture conditions in the
root zone (e.g. Sridhar et al., 2008; Bolten et al., 2010; Carrão et al., 2016; Martínez-Fernández et al., 2016). The well-known
and widely-used Palmer Drought Severity Index (PDSI, Palmer, 1965) calculates a simple water budget based on monthly
values of precipitation and potential evapotranspiration, in combination with parameters that have been optimized to ensure
similar PDSI values correspond to similar impacts on vegetation and crop yield even in different climate conditions. The
development of high-resolution land surface models applied at continental scales now also allows to have a more physically-
based alternative to PDSI, which can account for local soil and vegetation properties. In other cases, ranked or standardized
in situ or remotely sensed soil moisture observations have been used directly as agricultural drought index (e.g. Peled et al.,
2010; Mozny et al., 2012; Crow et al., 2012). Helped by the readily available satellite observations of vegetation indices
like NDVI, EVI, SIF, fPAR, NIRv and VOD, other studies have been focusing on the use of these vegetation indices to
quantify agricultural drought (Anyamba and Tucker, 2012; Hu et al., 2019; Buitink et al., 2020). Similarly, Narasimhan and
Srinivasan (2005) developed two separate indices for agricultural drought monitoring: one focused on soil moisture, and the
other on evapotranspiration. The current definition of agricultural droughts, i.e. *a soil moisture deficit severe enough to hamper
vegetation growth*, thus does not allow for a single index that can describe the entire drought, i.e. both its cause (soil moisture
deficit) and its impact (hampered vegetation growth).

Whereas soil moisture and vegetation-based indices both aim to quantify agricultural drought, the relation between soil
moisture and vegetation is characterized by considerable complexity and nonlinearity. Although combined indices have been
proposed as a solution (Yurekli and Kurunc, 2006; Sivakumar et al., 2010; Sepulcre-Canto et al., 2012), it can be questioned
whether agricultural drought should be quantified by a single index. From the small scale to the continental scale, distinct water-
and energy limited soil moisture regimes can be identified (Denissen et al., 2020), with the relation between soil moisture
and evaporative fraction often being represented by a bilinear relation (Seneviratne et al., 2010). Above the so-called critical
moisture content, evapotranspiration and plant functioning will not be limited or affected by a lack of precipitation. In fact,
increased incoming solar radiation during drought periods can even enhance evapotranspiration (Teuling et al., 2013), leading
to positive anomalies in vegetation indices despite prolonged meteorological drought conditions (Jolly et al., 2005; Teuling
et al., 2006; Mastrotheodoros et al., 2020; Kowalska et al., 2020). Because of this duality in the drought impacts, the use of
the term *agricultural drought* is ambiguous, even more so as the term *drought* bears a negative connotation to it, though its
impacts are not necessarily negative. The threshold behaviour associated with "absolute" critical moisture content is clearly at
odds with the current dogma that drought and its impacts are expressed relative to mean conditions.





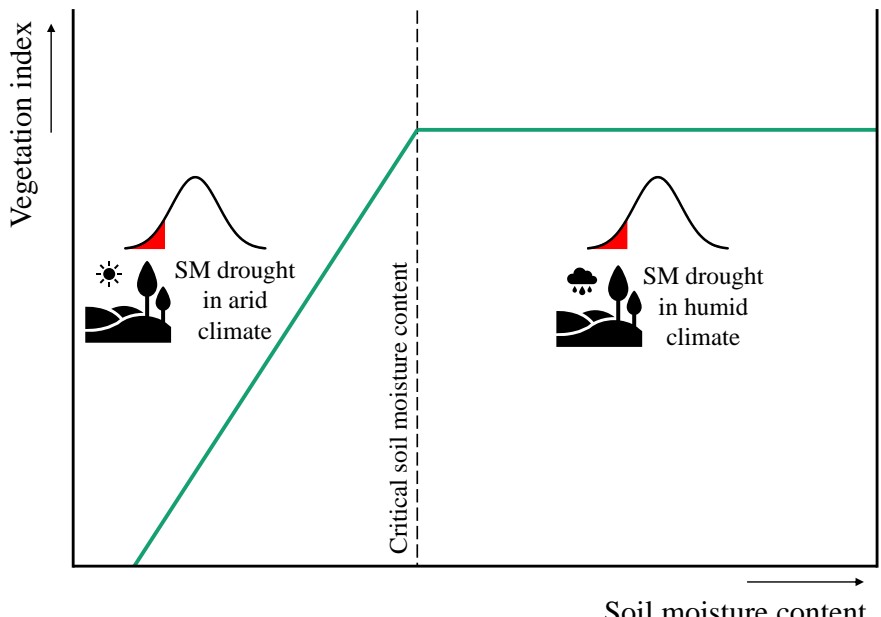

**Figure 1.** Illustration that different soil moisture climatologies can potentially mean very different things in terms of vegetation.

To address the issues surrounding the definition of agricultural drought, we aim to characterize the synchrony and similarity

between droughts in soil moisture and vegetation using readily available long-term gridded datasets of precipitation, vegetation

functioning, and soil moisture. Based on the concept of critical soil moisture, we hypothesize that the link between soil mois-

ture and vegetation droughts is more direct in the water-limited Mediterranean region, whereas a more complex behaviour is

expected in the more humid Northern Europe, bases on results from Peled et al. (2010) and Denissen et al. (2020). We investi-

gate the relation between soil moisture and vegetation drought for six widespread meteorological drought events that occurred

over the past two decades in Europe, including the severe 2003 and, more recent, 2018 events, that occurred in water- as well

as energy-limited regions. In addition, we critically evaluate the current practice of using soil moisture to predict observed

agricultural drought (i.e. vegetation impact).

## 2 Methods

### 2.1 Data

Daily remotely sensed soil moisture (SM) data was obtained from the ESA Climate Change Initiative soil moisture dataset

(ESA CCI SM v04.5, Gruber et al., 2017; Dorigo et al., 2017; Gruber et al., 2019), on a 0.25° resolution. The information

contained in satellite soil moisture data mainly contains surface soil moisture content, rather than root-zone soil moisture

content (Nicolai-Shaw et al., 2017), which has a more direct impact on vegetation performance. Regardless, remotely sensed

data was deemed most suitable for this study because of the long time period and large spatial scale of the analysis, and the





unavailability of root zone soil moisture measurements on such scales. Existing large scale root zone soil moisture datasets are either inferred from surface soil moisture, using land surface models (e.g. Crow and Tobin, 2018; Beaudoing et al., 2017; Houborg et al., 2012) , or using water balance models (e.g. Owe et al., 2008; Crow, 2012; Bauer-Marschallinger et al., 2018).

Monthly Normalized Difference Vegetation Index (NDVI) data was gathered from the MODIS dataset on a monthly timescale and a 0.05° resolution(MODIS MOD13C2, Tucker, 1979; Didan, 2015). Though MODIS vegetation indices are available on

a 16-day resolution, we opted for a monthly mean rather than a temporal composite, to have a more consistent sensing date throughout the dataset.

Soil moisture and vegetation data were resampled to the lowest spatial and temporal resolution of both datasets, resulting in a monthly 0.25° resolution. On this time scale, we assume that large scale patterns in both soil moisture and vegetation will remain similar, although lags between surface soil moisture and vegetation patterns are expected (Crow et al., 2012). The main

vegetation evolution occurs on a monthly timescale, not on a day-to-day basis, as soil moisture does. For comparison purposes the monthly timescale thus is more appropriate.

## 2.2 Drought selection

To study the relation between negative soil moisture and vegetation anomalies, growing seasons where significant precipitation deficits occurred were selected, based on the 6 month aggregated Standardized Precipitation Index (SPI6, derived from monthly

NASA GPM IMERG precipitation data, McKee et al., 1993; Huffman et al., 2019) in September of each year. As such, the SPI6 reflects the integrated precipitation deficit over the entire growing season. Interconnected pixels over relatively large areas, located in Europe (here 11° W–45° E, 35–72° N), with a strong precipitation deficit (SPI6 < −1) were chosen, resulting in the six selected seasons/areas as indicated in Fig. 2: the 2002 precipitation deficit over the Baltic states and north-western Russia (Rimkus et al., 2017), the 2005 event on the Iberian Peninsula (Sepulcre-Canto et al., 2012) and the infamous 2003, 2015 and

2018 events over central Europe (Ionita et al., 2017; Hanel et al., 2018; Buras et al., 2020). Because of the large North-South extent of the 2018 event, this event was split in two parts (hereafter referred to as *2018N* and *2018S*). Pixels in these selected areas were then used for further analysis, as discussed below.

## 2.3 Analysis

To allow for a fair comparison between anomalies of different variables, and to remove seasonal variations from the drought

definition, the data were normalized, by subtracting the long-term monthly mean from the SM/NDVI at each time step, and subsequently dividing by the long-term monthly standard deviation. This resulted in values between approximately −3 and +3, indicating negative and positive anomalies, respectively, that can be directly compared with SPI6. Other indices, such as the ESSMI (Carrão et al., 2016) for soil moisture data, or the VCI (Kogan, 1990) for NDVI data, are available and comparable to normalization, but a more general approach was adopted here to increase comparability of two different variables. We recognize

anomalies in SM (SMA) and NDVI (NDVIA) below −1 as pixels in drought. To account for seasonality in the variables, data for each month of the year were taken separately, and pixels with less than 7 data points were removed from the analysis. Both



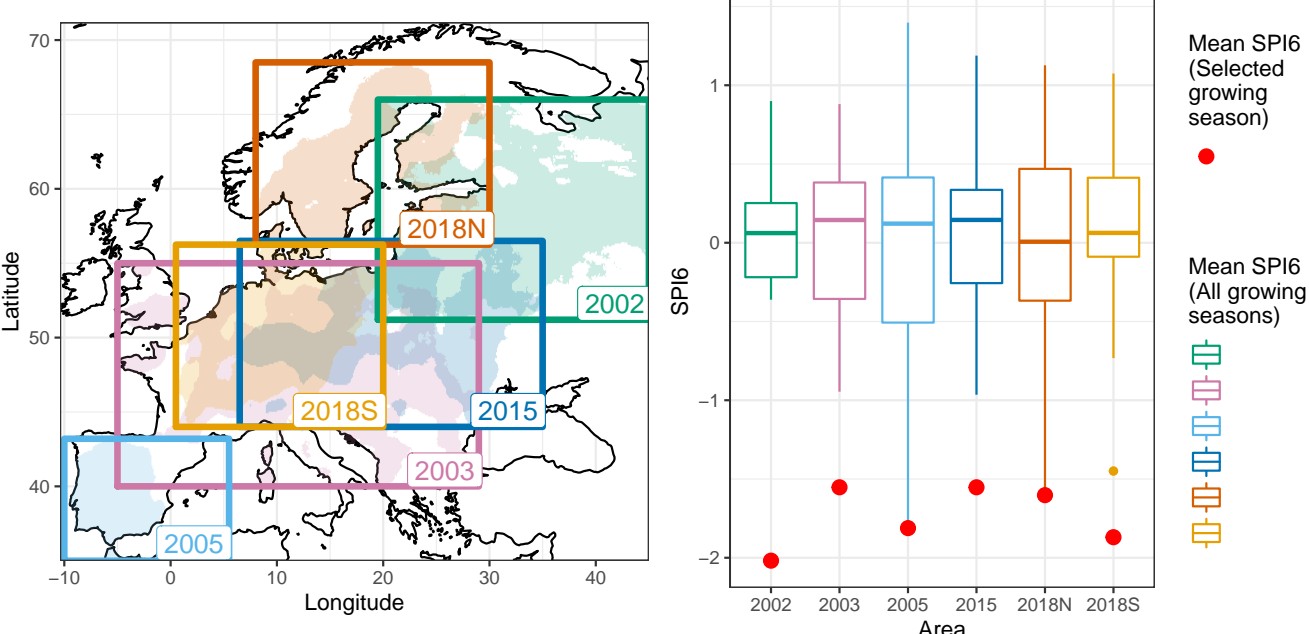

**Figure 2.** Properties of the selected summer droughts. Left: location and spatial extent, right: SPI6 over the selected growing season (red), compared to the distribution of SPI6 in the remaining growing seasons for the same region.

datasets have been extensively validated (e.g. Lahoz et al., 2018; Navarro et al., 2019), and, as such, a validation has not been conducted here.

After the data normalization, for each event, the fraction of the selected pixels in drought was determined for each variable.

Then, for each event and time step, the Pearson correlation between SMA and NDVIA was quantified. Though correlation is useful for an overview of similarity between two variables, it is not sensitive to bias or scale errors (Brier and Allen, 1951; Murphy and Epstein, 1989). Skill scores, on the other hand, give a more in-depth and well-rounded view on the use of SM as a predictor for agricultural impact. It should be noted that, because soil moisture indices are often used as a proxy for vegetation drought (e.g. Sridhar et al., 2008; Bolten et al., 2010; Carrão et al., 2016; Martínez-Fernández et al., 2016), predictions

using soil moisture drought are implicitly assumed to be skilful. Therefore, the number of Hits (H), Misses (M), Correct Rejections (CR) and False alarms (FA) were determined, and converted to five skill scores, each highlighting a different aspect of prediction accuracy. First, the Frequency Bias (FB) is given by:

$$FB = \frac{H+FA}{H+M}, \tag{1}$$





and expresses the difference between mean drought frequencies. Next, the Frequency of Hits (FOH) is a measure of discrimi-
nation, showing the fraction of forecasted vegetation droughts that were correct, which is given by:

$$\text{FOH} = \frac{\text{H}}{\text{H+FA}}. \tag{2}$$

The Frequency of Misses (FOM) is given by:

$$\text{FOM} = \frac{\text{M}}{\text{H+M}}, \tag{3}$$

and expresses the fraction of observed vegetation droughts that are incorrectly forecasted by the soil moisture anomaly. The
Hanssen-Kuipers score (HK,  Hanssen and Kuipers, 1965) measures the ability of the soil moisture drought to discriminate
between (or correctly classify) vegetation drought events and non-events:

$$\text{HK} = \frac{\text{H}}{\text{H+M}} - \frac{\text{FA}}{\text{FA+CR}}. \tag{4}$$

Lastly, the Odds Ratio (Stephenson, 2000, OR, ) is used to measure the strength of the association between soil moisture and
vegetation drought:

$$\text{OR} = \frac{\text{H} \cdot \text{CR}}{\text{FA} \cdot \text{M}} \tag{5}$$

We refer to Hogan and Mason (2011) for an overview of these, and more, skill scores, and their (dis-)advantages.

## 3  Results

A general check of the full data time series revealed that during each event, asynchronies between spatial patterns in soil
moisture and vegetation anomalies are widespread. Figure 3 serves as an illustration for these asynchronies, which occur in
all green and purple pixels (See Fig. S1-S5 for other events). Regionally more humid areas such as mountain ranges and high
latitude regions can easily be distinguished by their relatively low Pearson correlations between SMA and NDVIA (e.g. 0.10
in the Alps vs. 0.24 over the entire study area, see Fig. S6), in line with our hypothesis. Furthermore, correlations between the
anomalies were low in April and generally increased towards September, though in some areas, correlations peak in August.

Not all of the six studied events were equally affected by deficits in SM and/or NDVI. A comparison between drought extents
using the fraction of the area affected by a SM/NDVI drought is given in Fig. 4. The 2002, 2015, 2018N and 2018S events
are characterized by a clear overlap between the *NDVI* and *Both* lines, indicating that an area affected by an NDVI drought
also has a soil moisture drought. Interestingly, in 2003 and 2005, some negative vegetation anomalies occur in absence of a
SM drought. In these cases, vegetation growth was thus not obviously limited by current water content, but possibly by other
factors, such as energy, heat stress, antecedent low soil moisture conditions, or pests and diseases.
Since these events are located further south than most other selected events, energy limitations can be ruled out. Heat stress
could well have been the limiting factor for vegetation, as well as antecedent soil moisture anomalies, which had been negative
long before the growing season in 2003 and 2005 (Fig. S9), though NDVI anomalies (Fig. S11) were negative even before that,
indicating a poor state of vegetation all-together.





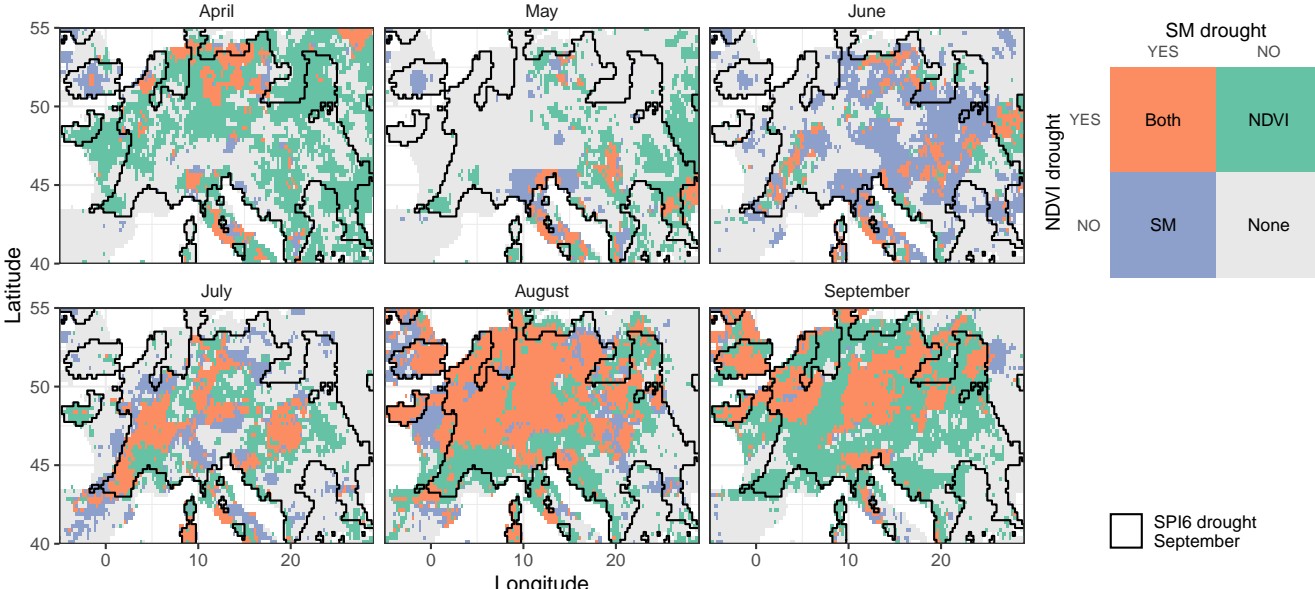

**Figure 3.** Synchrony between soil moisture and vegetation droughts during the 2003 growing season. Note the asynchronous development of soil moisture and vegetation drought, with soil moisture drought dominating in June, and vegetation in April and September. Similar figures for the other drought events are included in the Supplementary Material (Fig. S1-S5)

Figure 5 shows the severity of each drought event for both vegetation and soil moisture and the Pearson correlation between

NDVIA and SMA. Asynchrony between the two variables is visible in the irregular shape of the arrows, and the deviation of the linear regression from the 1:1 line. Generally, a delay can be distinguished between negative SMA and NDVIA values. This delay was expected, as discussed in Section 2.1. Interestingly, though, positive anomalies are more common in NDVIA than in SMA, showing that impacts of soil moisture droughts do not always show in the vegetation, and can sometimes even lead to opposite, i.e. positive, impacts in vegetation. High monthly correlations between SMA and NDVIA generally occur

late in the growing season, as shown by redder colours in Fig. 5. For example, in the 2002 event the NDVIA-SMA correlation increases from -0.04 in May to 0.57 in September, and correlations in the 2003 (2005, 2015, 2018N, 2018S) event peak in September (Apr, Aug, Jul, Jul), at 0.43 (0.34, 0.53, 0.38, 0.61). It seems that there is a general pattern that, when vegetation is energy limited, as is often the case the start of the growing season, NDVI remains largely unaffected by small anomalies in SM content, whereas under water limited conditions, which are more likely to occur near the end of the growing season, higher

correlations are found, consistent with results of Jolly et al. (2005). The southernmost 2005 event is the only event in which correlations peak early in the season, which can likely be related to the water-limited conditions that are likely to occur earlier in the Mediterranean than in the geographic locations of the other events.





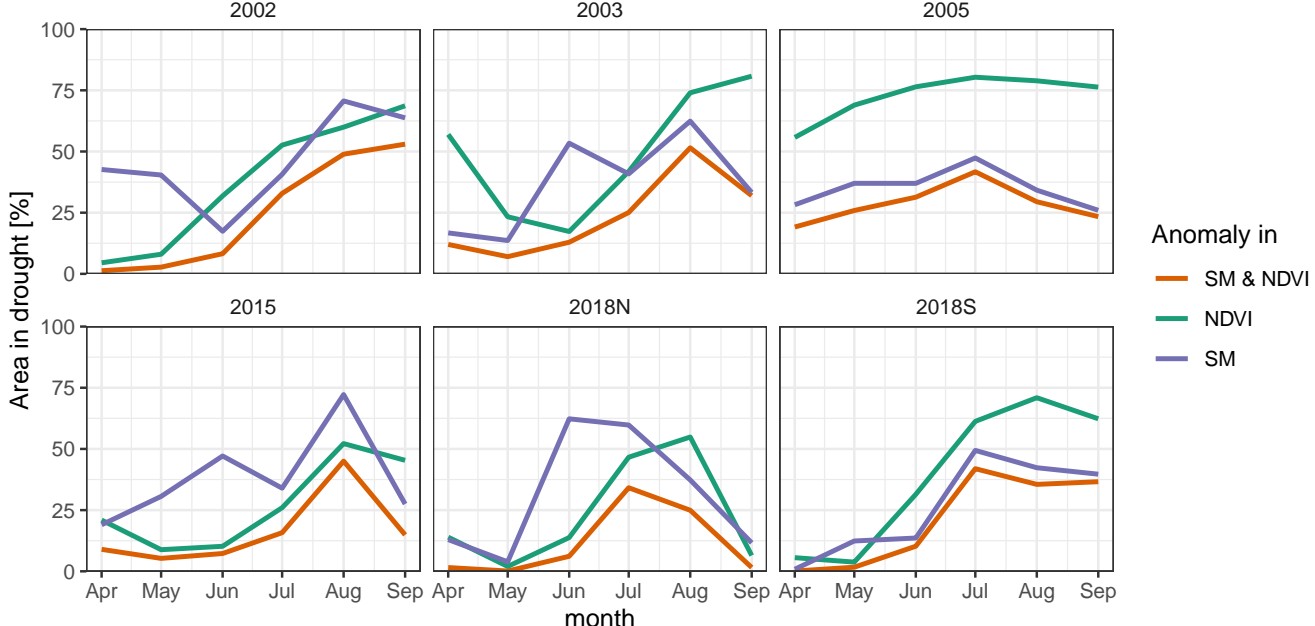

**Figure 4.** Growing-season evolution of percentage of area in drought. Panels show the six events, where NDVI (green) and SM (purple) pixels in drought (defined as an anomaly $< -1$) are shown separately, as well as the percentage of pixels affected by droughts in both variables simultaneously (orange).

Given the clear asynchrony and discrepancy in soil moisture and vegetation under water-limited conditions, it is relevant to question how well soil moisture-based indices, such as the widely-used SSMI and PDSI, perform when targeting to quantify

vegetation drought. The skill scores of agricultural drought impacts, as reflected in NDVI and as predicted using SMA, is shown in Fig. 6. From the low density of lines in the parts of the skill score plots shaded green, it is clear that the overall skill is rather low. Moreover, similar to the Pearson correlation, skill scores generally increase in August, though we expect the usefulness of end-of-season NDVIA prediction to be limited for agricultural purposes. Overforecasting, i.e. when more droughts are forecasted using soil moisture than there are droughts observed in vegetation, as seen in a FB > 1, generally

occurs in the beginning of the growing season, whereas underforecasting (FB < 1) occurs near the end of the growing season. The respective in- and decrease in FOH and FOM show the result of the changing frequency bias. The HK, showing the accuracy of events minus the accuracy of non-events, is rather stable throughout the growing season, though it peaks in the second half, just as the OR, which shows the number of correct forecasts. None of the drought events stand out in all of the skill scores. A sensitivity analysis showed that different thresholds for the drought selection and skill scores did not substantially

change the results.





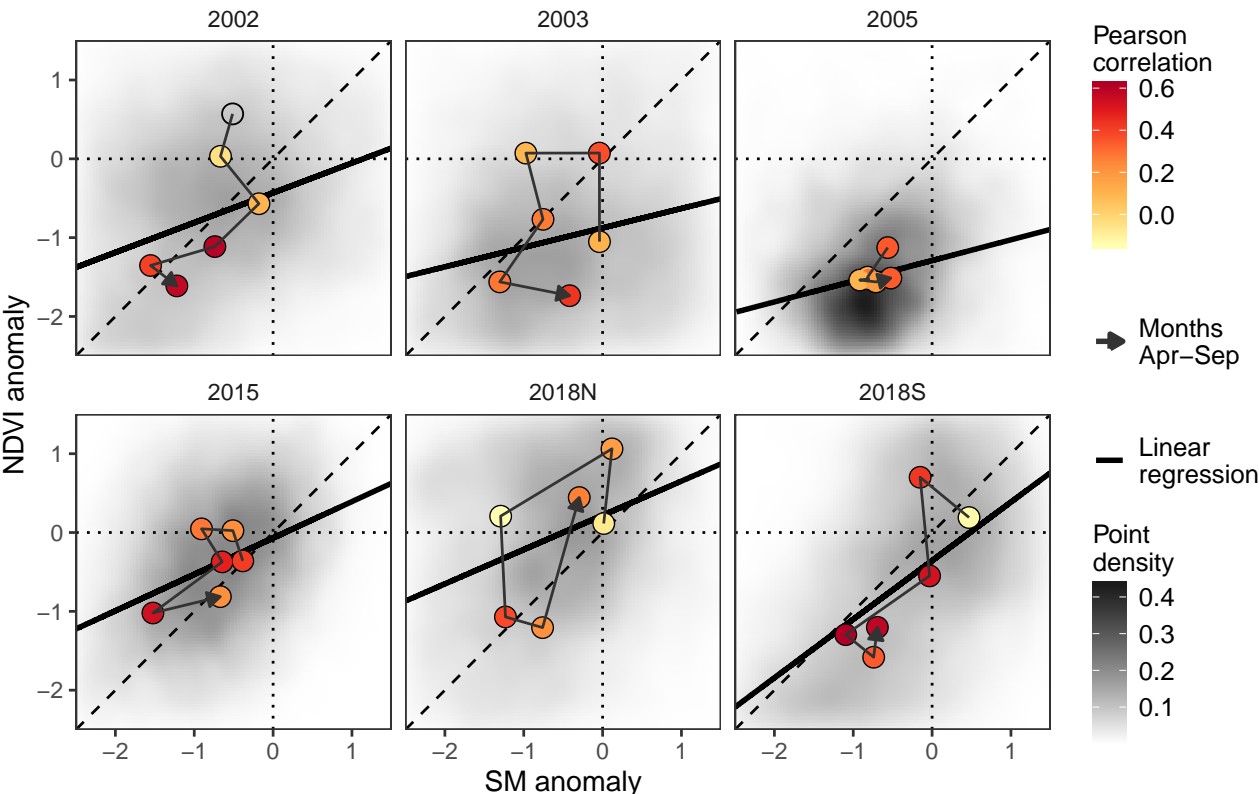

**Figure 5.** Relation between negative soil moisture and vegetation anomalies. Panels show the six drought events, with drought defined as anomaly $< -1$. Centroids of each month are chronologically connected with an arrow, and shaded by the Pearson correlation in that month, if $p \leq 0.05$.

## 4 Discussion

The complexity of agricultural droughts is not a local or regional issue, but a global one, and thus should be considered that way. While this study was performed over the European continent, it covers a range of climates found around the globe: from arid regions in the Mediterranean to boreal regions in northern Scandinavia. It is therefore expected that the behaviour will

be similarly asynchronous in other regions. Limitations of this approach are on a local scale, rather than the global scale, due to the low spatial resolution of the analysis. Even though each dataset was carefully selected based on their length, spatial resolution and validation results over Europe, resulting in a selection of datasets best suited for this analysis, uncertainties are inherent to any type of data and results should therefore be interpreted with care. In complex landscapes, high-resolution information can sometimes reveal a range in anomalies, even containing contrasting signs, that is not visible at coarser scale



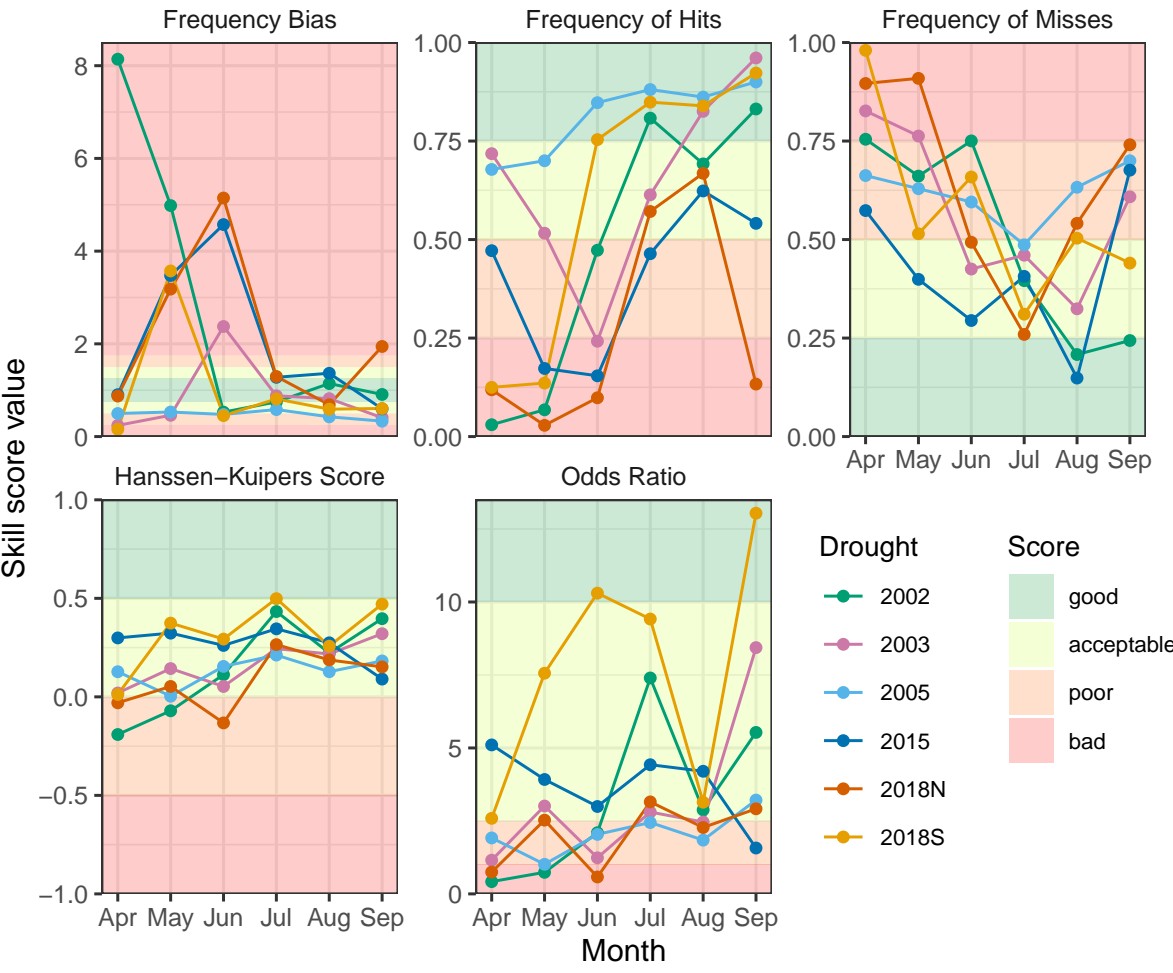

**Figure 6.** Skill scores for soil moisture drought as a proxy for vegetation drought. Background colours indicate the quality of the skill scores (see Methods for an, and the lines show different drought events.





Buitink et al. (2019). The normalizing of soil moisture data in this study can be criticized, because soil moisture data can show bimodality (Teuling et al., 2005; Vilasa et al., 2017). In addition, a dataset length of 18 years can be considered short compared to a traditional 30-year reference period, as recommended by the WMO (2017). On the other hand, uncertainties due to areal properties are decreased, because pixel values are compared to other values of the exact same pixel, while the resulting anomalies can easily be compared to other pixel values. This, next to the possibility to fairly compare different variables, led to

the decision to use a standard normalisation for both vegetation and soil moisture data, regardless of this method's limitations.

In this research we used available long-term satellite records of soil moisture and NDVI. Whereas current satellite soil moisture products are limited to the top few cm, a soil moisture drought assessment is ideally based on observations over the entire root zone. However, such observations are currently only available in several regional-scale observation networks (Mittelbach et al., 2011). Besides NDVI, numerous other products exist that reflect vegetation water status and/or productivity.

These include other indices based on optical (NIR, RED and BLUE) imagery (e.g. NIRv, EVI, etc.) or on microwave data (e.g. VOD). Though each of these different indices might give slightly different results, their application should not affect the fundamentally different response of soil moisture and vegetation to drought.

The inherently complex and nonlinear relation between soil moisture and vegetation status has important implications for drought monitoring, where traditionally a distinction is made between meteorological, agricultural, and hydrological drought

events. Whereas traditionally soil moisture has been used to indicate agricultural drought, our research highlights that a distinction is necessary between soil moisture drought (reflecting water status) and negative anomalies in vegetation (reflecting the impact of the drought on vegetation). This is particularly true when evaluating droughts across climate zones. The distinction between soil moisture drought and anomalies in vegetation is important, because shorter soil moisture droughts can even have a positive rather than negative impact on productivity, risking misclassification of drought events and false drought alarms.

## 5  Conclusions and outlook

Agricultural droughts are generally quantified using soil moisture anomalies, but our results show that clear asynchrony and discrepancies exists between these anomalies and their effects on vegetation. Occasionally, negative anomalies in soil moisture even lead to positive anomalies in vegetation. In some of the studied events, anomalies in vegetation could not be attributed to a soil moisture deficit alone. Though the asynchrony of soil moisture and vegetation deficits is not a novel finding (e.g. Crow

et al., 2012), the combined definition of agricultural droughts is still being used. To overcome this duality in the definition of agricultural droughts, and to prevent false drought alarms, drought monitoring and prediction may benefit from a move away from the combined term *agricultural drought*, that can lead to mixing up of soil moisture and vegetation effects, towards two separate terms: soil moisture drought and vegetation anomalies, each with their own indices and use in drought monitoring and forecasting.



*Data availability.* Precipitation data were recovered freely from the NASA Global Precipitation Measurement (GPM) Integrated Multi-satellitE Retrievals for GPM (IMERG) "Final run" P data, available from 2000 to present upon registration via https://disc.gsfc.nasa.gov/datasets/GPM_3IMERGM_06/summary. Soil moisture data, here used from 2000 up to and including 2018, are freely available from 1978 to present from the ESA Climate Change Initiative (ESA CCI SM v04.5), after registration at https://www.esa-soilmoisture-cci.org/. Monthly NDVI data were recovered from the MODIS MOD13C2 product and are available for free on NASA's LP DAAC website (https://lpdaac.

usgs.gov/products/mod13c2v006/).

*Author contributions.* All authors contributed to the design of the research. T.C.v.H. planned and carried out the simulations and took the lead in writing the manuscript. All authors contributed to the interpretation of the results and provided critical feedback and helped shape the research, analysis and manuscript.

*Competing interests.* The authors declare that they have no conflict of interest.

*Acknowledgements.* This project was supported by the Fonds National de la Recherche Luxembourg (FNR) (PRIDE15/10623093 – HYDRO-CSI).



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
