# Peer review of "Ambiguous agricultural drought: characterising soil moisture and vegetation droughts in Europe from earth observation"

_Hydrology and Earth System Sciences, 2020_

## Referee Comment (RC1) · Anonymous Referee #1 · 26 Jan 2021

The paper describes an investigation into the effect of soil moisture on agricultural droughts. It argues for a distinction between soil moisture drought and anomalies in vegetation as different responses of vegetation to soil moisture anomalies are observed. Using monthly standardized CCI SM and NDVI anomalies the correlation and skill scores between SM and NDVI are calculated for major European drought events. The analysis shows that depending on the soil moisture availability, i.e. water limited regions or other regions, SM and NDVI are stronger or weaker correlated and SM is a stronger or weaker predictor for NDVI anomalies.

I have difficulties to see the novelty of this research and I also have some issues with

lack of analysis and discussion of results. First of all, the current study seems to confirm results of previous studies using slightly different metrics but does not introduce any new findings. To name a few (and some of these papers are mentioned in the introduction, and there are more studies available): In 2010 Peled et al. published in HESS on the relation between SM, drought indices and NDVI. Stating that in northern areas there is a lower correlation between SM and NDVI, due to the strong influence of temperature on vegetation. In addition, they argue that correlations are stronger when soil moisture stress is higher. Over Australia Chen et al. (2014) have found similar results, demonstrating increasing correlation between satellite-based SM and NDVI with increasing soil moisture stress, both in space and time. One of the most comprehensive studies, from Nicolai-Shaw et al. (2017) showed the difference in vegetation response to drought for different regions, arguing that over regions such as Northern and Central Europe soil moisture deficits are not sufficient to limit ET or vegetation activity as here vegetation is radiation driven. The positive response of vegetation to dry and hot conditions has also been shown by Zscheischler et al. (2015). An overview of studies on interaction between CCI SM and vegetation is also given in Dorigo et al., (2017).

All these previously mentioned studies have already demonstrated the different response of vegetation to soil moisture anomalies. What these studies also all stated is the lagged response of NDVI to SM anomalies, which most commonly was found to be one month. What most of these studies also have addressed is the different response for different land cover types. Chen et al. (2014), Nicolai-Shaw et al. (2017), Peled et al., (2010), Liu et al. (2017), McNally et al., (2016) showed a different response in NDVI to SM anomalies for densely vegetated areas such as forests, compared to less dense vegetated areas such as grass- and croplands. I am missing the analysis on lags and land cover in the current study. As there has been no additional analysis on the lag between SM and NDVI anomalies I am left with the question if not taking into account the lagged response of NDVI can explain the low skill scores? It would be interesting to see how skill scores change with introducing a lag between SM and NDVI. Furthermore, the different land covers in Europe have not been addressed in this study. It is absolutely necessary to include land cover in the discussion of the results, as so many studies have already shown the different response. In Figure 4 it can also be seen that for some droughts the area in drought for SM seems to be preceding NDVI area in drought. This should be explained, and could it be related with the fraction of forest in the total area investigated per drought?

A more technical question on the data is how and if you masked for frozen soils and snow cover? The 2018N area goes up to ∼69°N, the 2002 area up to ∼66°N. You also use observations starting from April. In these regions there is a possibility that soils will be frozen or snow cover is still there. How did you mask for this? In Figure 4 in the 2002 drought there is a larger drought area in SM than in NDVI. As frozen soils and snow cover can lead to low backscatter and high Tb, this could lead to erroneously low SM values and possibly explain the larger drought area in SM in April and May for the 2002 drought? Another minor question, was the combined, active or passive dataset used?

Chen, T., R. A. M. de Jeu, Y. Y. Liu, G. R. van der Werf, and A. J. Dolman. 2014. "Using Satellite Based Soil Moisture to Quantify the Water Driven Variability in NDVI: A Case Study over Mainland Australia." Remote Sensing of Environment 140 (January): 330–38. https://doi.org/10.1016/j.rse.2013.08.022.

Dorigo, Wouter, Wolfgang Wagner, Clement Albergel, Franziska Albrecht, Gianpaolo Balsamo, Luca Brocca, Daniel Chung, Martin Ertl, Matthias Forkel, and Alexander Gruber. 2017. "ESA CCI Soil Moisture for Improved Earth System Understanding: State-of-the Art and Future Directions." Remote Sensing of Environment 203: 185–215.

Liu, N., R. J. Harper, B. Dell, S. Liu, and Z. Yu. 2017. "Vegetation Dynamics and Rainfall Sensitivity for Different Vegetation Types of the Australian Continent in the Dry Period 2002–2010." Ecohydrology 10 (2): e1811. https://doi.org/10.1002/eco.1811.

McNally, Amy, Shraddhanand Shukla, Kristi R. Arsenault, Shugong Wang, Christa D. Peters-Lidard, and James P. Verdin. 2016. "Evaluating ESA CCI Soil Moisture in East Africa." International Journal of Applied Earth Observation and Geoinformation, Advances in the Validation and Application of Remotely Sensed Soil Moisture - Part 2, 48 (June): 96–109. https://doi.org/10.1016/j.jag.2016.01.001.

Nicolai-Shaw, Nadine, Jakob Zscheischler, Martin Hirschi, Lukas Gudmundsson, and Sonia I. Seneviratne. 2017. "A Drought Event Composite Analysis Using Satellite Remote-Sensing Based Soil Moisture." Remote Sensing of Environment, Earth Observation of Essential Climate Variables, 203 (December): 216–25. https://doi.org/10.1016/j.rse.2017.06.014.

Peled, E., E. Dutra, P. Viterbo, and A. Angert. 2010. "Technical Note: Comparing and Ranking Soil Drought Indices Performance over Europe, through Remote-Sensing of Vegetation." Hydrology and Earth System Sciences 14 (2): 271–77. https://doi.org/10.5194/hess-14-271-2010.

Zscheischler, Jakob, René Orth, and Sonia I. Seneviratne. 2015. "A Submonthly Database for Detecting Changes in Vegetation-Atmosphere Coupling." Geophysical Research Letters 42 (22): 9816–24. https://doi.org/10.1002/2015GL066563.
* * *

---

## Referee Comment (RC2) · Anonymous Referee #2 · 1 Feb 2021

Summary:

This paper investigates soil moisture and vegetation anomalies during major drought events in Europe in the past two decades. The authors found asynchronies and discrepancies between negative near-surface soil moisture anomalies and vegetation (NDVI) anomalies, including a delay between the onsets of the two. In some cases, positive anomalies in NDVI were observed when anomalies in soil moisture were negative. The results highlights the importance of distinguishing between the terms soil moisture drought and anomalies in vegetation in drought monitoring and forecasting.

The analyses are related, and of good quality, including both a visualization and a quantification of the ability of soil moisture anomalies to predict concurrent NDVI anomalies over the growing season for the selected drought events. The manuscript is well written, and has a clear topic and conclusion. In my opinion, it is relevant within the scope of HESS. However, several necessary revisions are needed. My comments for improvements are listed below. P refers to page number, and L refers to line numbers in the manuscript.

Major comments:

The use of the term agricultural drought:

On P1L21-22 you refer to three main types of natural drought, i.e. meteorological, hydrological and agricultural drought. I want to bring to your attention that it has become more common in recent decades to use the term "soil moisture drought" instead of "agricultural drought" unless referring to the impact of soil moisture drought on a specific crop (yield), (e.g. https://doi.org/10.1002/wat2.1085, https://doi-org.ezproxy.uio.no/10.1038/491338a), in line with what you recommend in the manuscript.

Secondly, the term "agricultural drought" is commonly defined in relation to agricultural productivity and crop yield (e.g. https://doi.org/10.1016/j.jag.2015.06.011, https://drought.unl.edu/Education/DroughtIn-depth/TypesofDrought.aspx) and not to vegetation growth in general (P2L28). It is the specification of agricultural impacts that links the definition directly to food production and irrigation (P2L28-29). In the main analysis of the present study, I am missing a discrimination between crop, vegetation or land cover types. Accordingly, I recommend you to comment on the use of NDVI across Europe regardless of agriculture, land cover or vegetation type, to quantify agricultural drought. Adding a few lines in the introduction (e.g. after P2L39-41) reviewing various indices (and in particular NDVI) used to quantify agriculture drought, will support your choice of dataset. I would also recommend you to comment on the choice of the fixed growing season (note for example that northernmost parts of your

study region typically have a later onset of a frost-free growing season). Lastly, it is currently unclear to me whether you argue for moving away completely from the usage of the term "agricultural drought" and rather use "soil moisture drought" and "anomalies in vegetation", or whether "agricultural drought" can be used as a term for vegetation anomalies.

I would recommend you to include some sentences in the introduction presenting different practices in how to define and quantify agricultural drought. Addressing the abovementioned comments directly in the manuscript, could help placing your study in the broader scientific context and clarify the novelty of your concluding message.

The use of remotely sensed soil moisture data:

As mentioned on P3L72, the soil moisture dataset used in the study, mainly contains near-surface soil moisture content, rather than root-zone soil moisture content. You have justified this choice; however, the potential limitations of using a near-surface dataset are under-communicated. In the introduction you clearly introduce that agricultural drought have traditionally been quantified based on soil moisture conditions in the root zone. Near-surface soil moisture is typically quicker-responding and more directly related to the meteorological conditions as compared to root-zone soil-moisture. Notable when only considering the precipitation anomaly aspect of a meteorological drought, as in the present study. Larger asynchronies and discrepancies between soil moisture and vegetation anomalies may therefore be expected when using near-surface as compared to root-zone soil moisture, hence potentially giving the present study a too 'optimistic' conclusion (i.e. overestimating the asynchrony).

I recommend to include a paragraph in the introduction about the benefits/drawback of the different soil moisture datasets, and differences between near-surface soil moisture and root-zone soil moisture, as well as a paragraph in the discussion about the related uncertainties (e.g. too what degree/where/when could you expect similar results if you had used root-zone soil moisture). The discussion part could be added after the

sentences on P11L191-194. I would also recommend being clearer about the usage of near-surface soil moisture data in the abstract, aim and conclusion.

Novelty:

On P11L209 you acknowledge that the asynchrony of soil moisture and vegetation deficits is not a novel finding, referring to Crow et al. 2012. Related to my first comment, it is already common to make a distinction between soil moisture drought and vegetation anomalies. For example, the European Drought Observatory (https://edo.jrc.ec.europa.eu/edov2/php/index.php?id=1000) separate between rainfall deficit, soil moisture deficit and vegetation stress. I recommend acknowledging when/where it is already common to make this distinction. In addition, including a small review of previous findings related to the asynchrony of soil moisture and vegetation in the introduction, and how that compares to the objective of the present study, is needed to clarify in what way(s) your study is contributing with new knowledge.

Concepts consistency:

Please be consistent in the usage of (or be clear about when using the different) main concepts in the manuscript. Now, it is unclear when/whether different names refer to the same. For example, the concepts "vegetation anomaly", "vegetation drought", "NDVI drought", "agricultural drought", "vegetation impact", "agricultural drought impact" and "NDVI". Also "SPI6", "summer drought", "SPI6 over the selected growing season", "SPI6 drought September". The clarification would also benefit from an explicit distinction between (what is regarded) drought and (what is regarded) impact in your analysis. This is particularly the case for vegetation, which is sometime referred to as (agricultural-, vegetation-, NDVI-) drought and sometimes as (vegetation-, agricultural-, agricultural drought-) impact.

Critical soil moisture:

A more thorough introduction to the concept critical moisture content and energy vs water limited regions, could help setting the scene for the arguments later in the section (end of P2 and P3), as well as in the discussion of the results. For example, include the definition, expanding from that it varies to how it varies (e.g. over the region and over time, and between land cover, soil, vegetation types), and the main finding of Peled et al. (2010) and Denissen et al. (2020) on which you base your hypothesis. I would also recommend you to add a few sentences elaborating the discussion of critical soil moisture (and energy vs water limited) in relation to your findings, to follow up the hypothesis you introduce. A more thorough introduction and discussion of these complexities can further strengthen your argument for why one should make a distinction between soil moisture drought and vegetation anomalies.

Results and Discussion:

Results and Discussion are structured as two separate chapters; however, the distinction between results and discussion is not clear in the Results chapter. I recommend clarifying for the reader where the discussion of the results are found by moving the discussions of the results to the Discussion chapter. In addition, I recommend including subsections with subtitles to make the structure clear for the reader. In particular, including a subsection discussing the hypothesis in relation to your findings, and a subsection reflecting P3L66-67 ("critically evaluate the current practice...").

Period:

The period(s) used are lacking in the manuscript, including in the Data Section, in the anomaly and drought calculations and selection, and in the figure captions in the Results chapter. Please include the periods of the datasets in the Data Section, and the chosen period(s) for all relevant analyses (e.g. choice of reference period for SPI6, soil moisture and vegetation anomaly, and period for which selection of drought events was undertaken). If they vary (e.g. use of different reference period), please explain why and discuss possible implication this may have for the results.

Figures:

The visualization in the Figures are generally clear and appealing. However, the names given in the Figures are not always in line with the names given in the figure caption, and the captions are not always clear or complete. In each figure caption, I recommend explaining all abbreviations and concepts in the corresponding figure to make them understandable for the reader without having to search through the main body text. Some examples include abbreviation "SM" (e.g. Fig. 1), and concepts "summer" (e.g. Fig. 2) and "growing season" (e.g. Fig. 2).

Minor comments:

P1L15: Based on the argument and references, I am wondering whether you mean "drought" instead of "impact of drought" (as is written)? The way it is formulated now the link between climate change/enhanced land-atmosphere feedback and more severe drought impacts (i.e. changes in drought characteristics) is missing. Please consider rephrasing to include this 'link' (e.g. more in line with the first sentence of the abstract).

P2L32-35: Please add recent references using PDSI to study agricultural drought over your domain (Europe) to support your statement. To my knowledge, this index is mainly used in America.

P2L53: Clarify which drought type (e.g. meteorological drought) in "during drought periods", to avoid confusion. Note that "drought" without specification is ambiguous, especially when dealing with different types of drought as is the case for this study. I recommend to specify the drought type here and throughout.

P3L60: Adding "historical", "remotely sensed data", and/or the period could clarify already in the introduction that you are using historical/remotely sensed datasets (as opposed to e.g. model runs, projections, reanalysis). Adding the period could also clarify the indistinct concept "long-term". As stated on P11L186, 18 years can be considered short.

P3L61-63: Change the wording to avoid "Based on" and "bases on" in the same sentence.

P3L72-73: Grammatically unclear whether the "which" statement refers to surface soil moisture content or root-zone soil moisture content.

P3 Fig. 1: Clarify the message of the figure by elaborating in the caption what the graph and symbols illustrate. In addition, the figure will be more clear by being consistent in the wording; e.g. does "soil moisture climatologies" refer to the binary distinction between "SM drought in arid climate" and "SM drought in humid climate", or the continuous x-axis of "Soil moisture content" in the figure please provide a more precise explanation of what is meant by "Vegetation index" in the y-axis and "Vegetation" in the caption (which aspect of vegetation is meant).

P3L68: Consider changing the title to "Data and methods".

P4L82: Be more precise about which resampling method is used (both spatial and temporal) to make it reproducible.

P4L84: Please clarify why lags are expected between soil moisture and vegetation patterns when a monthly time scale is applied. Why is this lag not accounted for in your analysis (e.g. skill scores of the soil moisture anomaly's ability to predict the next month's NDVI anomaly)?

P4L85: Consider adding "near-surface" before soil moisture, or adding a reference to underline the statement. I would expect deeper soil moisture to have a slower main evolution.

P4L89-90: Please elaborate how the SPI6 is calculated in this study to make it reproducible. There are multiple ways to calculate SPI6, depending on the choice of distribution used to fit the data, handling of zero precipitation and choice of reference period. All these choices can affect the regions of September SPI6<-1.

P4L89-90: Why is the dataset used for SPI6 calculation not introduced in the Data Section?

P4L91: Include the months comprising the growing season in parenthesis in the end of the sentence. Also, I would assume the growing season starts even earlier in parts of the Mediterranean, and later in northern Sweden, thus the wording "entire growing season" does not apply everywhere? Consider changing the wording from "entire" to "a typical" or similar.

P4L92: I disagree with the phrasing "strong precipitation deficit" for SPI values smaller than -1, and refer to Table II in https://doi.org/10.1002/joc.846 for categories of SPI values. Here "moderate drought" refer to -1.49<SPI<-1, i.e. the major proportion of SPI<-1 events. A phrasing such as "moderate to extreme precipitation deficit" would be more in line with this categorization.

P4L100-101: Specify that this was done in a grid-wise manner. Specify what period "long-term" refers to.

P4L105: As for P2L53, I recommend to specify the drought types, e.g. "We recognize anomalies in SM (SMA) and NDVI (NDVIA) below −1 as pixels in soil moisture drought and vegetation drought, respectively."

P4L105: Consider including the reason for your 'moderate' drought (ref. comment P4L92) threshold choice.

P5L107: "Both datasets" refer to soil moisture and NDVI, whereas the references given are for soil moisture and precipitation.

P5L109: Please specify whether it is the fraction of all cells in the European domain, or only of the cells in the spatial extent of the individual drought events. This also applies for caption of Fig. 4.

P5L110: Consider adding "for each grid cell" for clarification.

P5L115-117: Specify for which variables/metrics the number of Hits (H), Misses (M), Correct Rejections (CR) and False alarms (FA) were determined. I assume it is the SMA<-1 vs NDVIA<-1, but this is not clearly stated.

P6L131: Consider whether some of the (dis-)advantages are relevant for the present study, and include them if they are.

P6L135-136: Here, you support your hypothesis by comparing the Alps with the remaining region. Could you elaborate? Specifically, I am wondering whether the low correlation is due to the Alps being energy-limited as you indicate, or whether it is (also/partly) due to the Alps having a high local heterogeneity in topography, soil moisture and (other) vegetation types as compared to the remaining region. Snow may also be a factor affecting the results in this region. The elaboration could be included in a paragraph discussing the hypothesis up against the results.

P7L152-154: Please rephrase to clarify the usage of "impacts". More specifically, "impacts of soil moisture droughts do not always show in the vegetation" indicate that there are impacts of soil moisture regardless of whether it is shown in the vegetation (for which a natural next step is to clarify which impacts), which I believe is not what you intended to communicate here.

P7L154: Be careful with the use of "lead to" in the case of presenting results of overlapping occurrences. It could be other variables that lead to the positive vegetation anomalies. Comment applies also for P11L208.

P8L165: Related to comment P5L115-117: Is it not NDVIA<-1 as predicted using SMA<-1, instead of NDVI as predicted using SMA?

P8L174: To clarify for the reader the sensitivity analysis you have performed, please specify the sensitivity analysis performed, and potential changes (though not substantial) seen e.g. when decreasing the drought threshold.

P9 Fig. 5: Please rephrase "negative soil moisture", which does not exist per definition. Also, note that "negative" can be a misleading word in the caption as the SM anomalies are not always negative in the Figure. Secondly, refer to the names in the Figure when explaining the Figure in the caption (e.g. "vegetation (NDVI) anomalies", and what

points the point density refer to). Thirdly, clarify what drought index is used in the definition in the second sentence (only "drought" is ambiguous), or refer to Figure 2.

P9L181-182: Adding a sentence in the Data section about the reasoning behind the selection of the NDVI dataset would give basis to the statement given here.

P10: Incomplete sentence in the caption of Fig. 6.

P11L185: "Buitink et al. (2019)" should either be in parenthesis, or is not a continuation of previous line on page 9.

P11L194-197: Why did you choose to use NDVI and not the other products mentioned here (I assume it relates to the careful selection of datasets mentioned on P9L181)? Consider including the answer in the manuscript.

P11L196-197: Please back up statement with relevant arguments and references.

---

## Author Comment (AC1) · 1 Feb 2021

First of all, we wish to thank the reviewer for reading our manuscript and for taking the time to provide us with insightful comments on our manuscript. In this document we will comment on the reviewer's main concerns.

The reviewer states that (s)he has difficulties to see the novelty of this research. With our study, we aim to contribute to the debate on the use drought indices (for agricultural drought in particular), and how Earth observation products can be used for this. We did not aim to provide an analysis on the (cor)relation between vegetation and soil moisture from a more process-based point-of-view, since, as the reviewer correctly

points out, many other studies have already looked at this. While the other references are relevant and we are more than happy to discuss them in a revised version of the manuscript, it seems that the Peled et al., (2010) paper is the most relevant to the context of our work. We note, however, several important differences, for instance its focus on seasonal rather than monthly timescales, its use of modelled rather than observed soil moisture, its focus on correlations rather than discrete drought events, and the now more historical period (1982–2002) which excludes many of the well-documented drought events that have occurred over the past 2 decades (including 2003 and 2018) that are covered in our study. We think that the main contribution of our work is providing an answer to the question how well one type of agricultural drought can be predicted by the other type from Earth observation data – an issue that we believe is novel, given that it is rare to see studies that make an explicit distinction between soil moisture and vegetation when discussing agricultural drought. The IPCC, for instance, defines 'Agricultural drought' as moisture deficits in the topmost one metre or so of soil (the root zone) that impact crops, without acknowledging that this relation is ambiguous when vegetation and soil moisture are analysed jointly (as we do).

The second point the reviewer makes is that (s)he found there is a lack of analysis and discussion of results. More specifically, the reviewer missed the analysis on lags and land cover in the current study. Our reason for not including this was that NDVI is used for real-time monitoring of agricultural drought in several national and European platforms. This data is used without lag, and hence in the framework of our research question (how should agricultural drought be defined?) we initially thought this was of less relevance. However the reviewers' comments have made us rethink this decision, and we believe an analysis of NDVI with a 1-month lag can provide useful insight because this is often assumed as a working hypothesis (as is confirmed by the reviewer). It should be noted that according to our hypothesis, whether the 1-month lag actually works might depend on the climate setting.

We repeated our original analysis, but now including a lag between soil moisture and

vegetation data was taken into account. Fig. 1 shows the results with this lag, as compared to the no-lag analysis shown in Figure 6 in the manuscript. Though the results are actually very comparable to the original analysis, it seems that in this case the skill scores are generally lower compared to the situation without lag. This indicates that a lag between the two variables does not explain the low skill scores obtained when attempting to predict one type of agricultural drought (soil moisture) by another (vegetation), thereby further strengthening our conclusion that soil moisture droughts should not be used as a proxy for vegetation drought.

Regarding the analysis based on different land covers, we refer to our supplementary material. There, we included several figures showing the NDVI and soil moisture anomalies for two different land covers (grassland vs. forest). We opted not to include these analyses in our main manuscript as there was a substantial difference between the amount of pixels of each land cover in the each drought event (Fig. 2). Robust results, comparing all of the studied drought events, could thus not be generated. What the figures in the Supplementary materials do show is that compared to the NDVI anomalies (NDVIA) in all pixels (Fig. S11), NDVIA in grassland pixels (Fig. S8) reacts much more strongly to a decrease in soil moisture than NDVIA in forested pixels (Fig. S7). One would therefore assume that the skill scores would indeed be higher over grassland pixels than over forested pixels.

To test this hypothesis, we repeated our original analysis for the two land cover types (as discussed in the supplementary material). Here, we include results of an analysis with an 80% threshold, meaning that pixels are included in the analysis only when at least 80% of its surface belongs to one of the two land cover types. The skill scores for grassland cover and forested cover are given in Fig. 3 and 4, respectively. These figures indeed show that there is a difference in skill when the two types of land covers are considered, as rightfully suggested by the reviewer. Results should be interpreted with care though, as the number of pixels considered per event differ strongly (Fig. 2). Regardless, we'd be happy to include these analyses in the manuscript or the

supplementary materials, dependent on the reviewer's preference.

The reviewer additionally comments on Figure 4 in the manuscript, where it can be seen that for some droughts, the area in drought for SM seems to precede the NDVI area in drought. That is indeed true, most notably for the 2015 and 2018N event. The suggestion of the reviewer that this is caused by the fraction of forest seems unlikely, based on the values given in Fig. 2. We would argue that these patterns, i.e. a soil moisture drought preceding a vegetation drought, are to be expected due to the lag that is common between soil moisture and vegetation anomalies, as shown, amongst others, in several papers cited by the reviewer (Chen et al., 2014; Nicolai-Shaw et al., 2017). In a possible revision of our paper, we will discuss this behaviour as well.

The reviewer finishes off with raising some questions regarding our analysis. The reviewer mentions that frozen soils and snow cover may have caused low soil moisture values in April and May for the 2002 drought, if not properly masked. Though we did not mask the soil moisture data ourselves, retrieval under such conditions is highly uncertain and therefore masked in the CCI combined soil moisture data set (Scanlon et al., 2020), which we used. Having said that, it has been found that there are still some issues with insufficient masking of snow or frozen soil conditions in this data set (van der Vliet et al., 2020). The low soil moisture values in April and May 2002 could thus indeed have been caused by low temperatures. We will include this in the manuscript.

References

Chen, T., de Jeu, R., Liu, Y., van der Werf, G., & Dolman, A. (2014, January). Using satellite based soil moisture to quantify the water driven variability in NDVI: A case study over mainland Australia. Remote Sensing of Environment,140, 330–338. doi: 10.1016/j.rse.2013.08.022

Nicolai-Shaw, N., Zscheischler, J., Hirschi, M., Gudmundsson, L., & Seneviratne, S. I. (2017). A drought event composite analysis using satellite remote-sensing based soil moisture. Remote Sensing of Environment,203, 216–225. doi:

10.1016/j.rse.2017.06.014

Peled, E., Dutra, E., Viterbo, P., & Angert, A. (2010). Technical Note: Comparing and ranking soil drought indices performance over Europe, through remote-sensing of vegetation. Hydrology and Earth System Sciences,14(2), 271–277. doi: 10.5194/hess-14-271-2010

Scanlon, T., Pasik, A., Dorigo, W., De Jeu, R. A. M., Hahn, S., van der Schalie, R., . . . Preimesberger, W. (2020) .ESA Climate Change Initiative Plus-Soil Moisture. Algorithm Theoretical Baseline Document (ATBD)(Tech. Rep. No. D2.1 Version 04.7). Austria: Earth Observation Data Centre for Water Resources Monitoring (EODC) GmbH.

van der Vliet, M., van der Schalie, R., Rodriguez-Fernandez, N., Colliander, A., de Jeu, R.,Preimesberger, W., . . . Dorigo, W. (2020). Reconciling Flagging Strategies for Multi-Sensor Satellite Soil Moisture Climate Data Records. Remote Sensing,12(20), 3439.(Number: 20) doi: 10.3390/rs12203439

———————————————————

[Figure]

Fig. 1. Skill scores for soil moisture drought at time t-1 as a proxy for vegetation drought at time t.

| Event | 80% | | | | 90% | | | |
|---|---|---|---|---|---|---|---|---|
| | Grassland | | Forest | | Grassland | | Forest | |
| | SM | NDVI | SM | NDVI | SM | NDVI | SM | NDVI |
| 2002 | 146 | 146 | 1060 | 1066 | 43 | 43 | 350 | 350 |
| 2003 | 730 | 844 | 51 | 232 | 300 | 335 | 5 | 49 |
| 2005 | 138 | 138 | 52 | 52 | 67 | 67 | 13 | 13 |
| 2015 | 553 | 560 | 116 | 122 | 193 | 196 | 25 | 27 |
| 2018N | 18 | 18 | 631 | 532 | 4 | 4 | 136 | 136 |
| 2018S | 355 | 355 | 52 | 52 | 100 | 100 | 5 | 5 |

**Fig. 2.** Number of pixels for each drought event when the different (80% or 90%) thresholds for two types of land cover are applied, as defined in the Supplementary information

**Fig. 3.** Skill scores over grassland pixels, as defined in the supplementary material. An 80%
threshold was used.

**Fig. 4.** Skill scores over forested pixels, as defined in the supplementary material. An 80% threshold was used.

[Figure]

---

## Referee Comment (RC3) · Rogier Westerhoff (Referee) · 5 Feb 2021

This paper discusses whether the term agricultural drought is ambiguous, according to the authors defined as 'a soil moisture deficit severe enough to hamper vegetation growth'. It compares soil moisture derived drought indicators with a vegetation derived indicator and ambiguities, e.g. a time lag in agricultural drought anomalies.

I would recommend rejection for this paper. I don't mean to say the findings are not useful, because they could form additional arguments to existing research. But to my knowledge this work could be specified as a 'discussion/review' . There are no novel findings, unless you would count the activity of comparison of CCI-SM with MODIS-

[Figure]

NDVI as a novel exercise. Also, the findings of time lag and differing correlations between soil moisture and vegetation derived indicators is also not new, as indicated by the authors who mention the Crow et al (2012) study.

Moreover, and probably more important, the pitch of the paper does not hold up well. The paper argues about a wrong/ambiguous definition of agricultural drought and there are a couple of issues with that argument throughout the paper:

1) The authors mention the current definition of agricultural drought as 'a soil moisture deficit severe enough to hamper vegetation growth'. Where does this definition come from? Please refer to a document. To my knowledge there are already quite a few different definitions of agricultural drought, as well as a variety of agricultural drought indicators. See other reviewers references, and add to that work of e.g. Cao et al (2019), Dalezios et al (2017), Heim et al (2002), Quiring et al (2003).

2) Given the authors' definition of 'a soil moisture deficit severe enough to hamper vegetation growth', the argument that it is ambiguous is not explained well enough (I did not understand at least). The authors claim that in certain soil moisture deficit NDVI shows actual increased growth. But that would not undermine the definition of 'a soil moisture deficit severe enough to hamper vegetation growth'. Maybe the authors have been looking at soil moisture deficits that were not enough to hamper vegetation growth?

3) A few other things need to be clarified. If it is known that the effect of vegetation lags the effect of soil moisture deficit, then the finding of increased growth with lower soil moisture deficit is highly uncertain. After all, the increased growth could be coming from a lagging soil moisture high before the deficit. The finding that correlation between the soil moisture and vegetation derived indicator is peaking at the end of the summer is probably mainly because the soil water deficit is lowest at the end of summer. So the authors are basically claiming that the 'soil moisture deficit is severe enough to hamper vegetation growth', which equals the author's definition of agricultural drought that they
claim was ambiguous in the first place.

Finally, other explanations of the differences found are not touched upon, such as: - groundwater table (e.g. a global version by by Fan et al, 2013, could have been used to identify root water uptake from shallow water tables) - differences in vegetation type; - influence of soil; - influence of fertiliser.

References: Quiring et al (2003): https://doi.org/10.1016/S0168-1923(03)00072-8 Cao et al (2019): doi:10.3390/rs11091066 Dalezios et al (2017): https://www.researchgate.net/publication/320689949_Agricultural_Drought_Indices_Combining_Crop_Climate_and_Soil_ Heim et al, 2002. https://doi.org/10.1175/1520-0477-83.8.1149 Fan et al. (2013). 10.1126/science.1229881

---

## Referee Comment (RC4) · Rogier Westerhoff (Referee) · 5 Feb 2021

Correction: on page C-2 I said 'because the soil water deficit is lowest at the end of summer'. Of course I meant 'because the soil water deficit is HIGHEST at the end of summer'.
* * *

---

## Author Comment (AC2) · 12 Feb 2021

First of all, we wish to thank the reviewer for reading our manuscript and for providing their insightful and detailed comments on our paper. In this document, we will address the comments of the reviewer. The major comments addressed are the use of agricultural droughts, the use of remotely sensed soil moisture data, novelty of the manuscript, consistency of concepts, critical soil moisture, results and discussion, periods, and the figures, each discussed in its own Section below.

1. The use of the term agricultural drought

[Figure]

The reviewer rightfully notes that it has become more common in recent decades to use the term "soil moisture drought" (SMD) instead of "agricultural drought" (AD). Indeed, the terms are often used correctly. Nonetheless, we found that using the term AD for denoting SMD is still rather common [e.g. Sridhar et al., 2008, Hao and AghaKouchak, 2013, Chakrabarti et al., 2014, Martínez-Fernández et al., 2015, 2016]. For that reason, we suggest to move away completely from the term agricultural drought towards soil moisture drought and vegetation drought, to avoid any ambiguity in drought research.

The reviewer also mentions that the term AD is commonly defined in relation to agricultural productivity and crop yield, rather than vegetation growth in general, as we wrote. We will adapt our phrasing in a newer version of the manuscript.

The next point the reviewer makes is that they miss a discrimination between crop, vegetation, and land cover types. In a new version of the manuscript, we will comment on the use of NDVI across Europe. We will additionally add figures in the appendix or supplementary material, including an analysis of skill scores split up by land cover, i.e. grassland/agricultural areas and forest (as shown in Figs. 1 and 2). We will also comment on the choice of the fixed growing season.

We agree that including such statements will improve the manuscript and help place the study in a broader scientific context.

2. The use of remotely sensed soil moisture data

The reviewer comments on the use of remotely sensed soil moisture data, which mainly give a view on the surface soil moisture, rather than the root zone soil moisture. We are indeed aware of this, but should be more elaborate on the disadvantages of this choice, and add an hypothesis on how our conclusions might have differed if large scale root zone soil moisture observations were available and used in this study. We will make the suggested changes in a newer version of our manuscript.

**3. Novelty**

Here, the reviewer makes a comment related to the one in Section 1, stating that it is already common to make a distinction between SMD and vegetation anomalies. In a newer version, we will more strongly address this and acknowledge cases where the distinction is made correctly. We will also expand our review on previous findings related to the asynchrony of soil moisture and vegetation in the introduction, as suggested by the reviewer.

**4. Concepts consistency**

The reviewer is absolutely right in pointing out that we should be consistent in our phrasing of important terms in our manuscript. We will choose one phrase per concept to be used consistently, and explain related terms and concepts in a newer version of the manuscript.

**5. Critical soil moisture**

Here, the reviewer gives a suggestion on how to improve our introduction and discussion by making use of the concept of critical soil moisture. We think that this is an excellent suggestion and will therefore incorporate this in a newer version of the manuscript.

**6. Results and Discussion**

The reviewer notes that there is no clear distinction between the results and the discussion. They suggest that we move the discussion away from the results section and into the discussion section. We agree that an improper distinction between these two sections may be confusing and we will therefore make sure to limit the discussion to the Discussion section in a newer version of the manuscript. Including subsections might help structure the discussion further and we will explore this suggestion when rewriting the Results and Discussion sections.

**7. Period**

The reviewer notes that the used periods are lacking in the manuscript, and more specifically in the Data section, in the anomaly and drought calculations and selection and in the figure captions in the Results chapter. We will include the used periods in the sections, as suggested by the reviewer.

8. Figures

We thank the reviewer for their positive feedback of the visualisation in the Figures. In a newer version of the manuscript we will explain all abbreviations and concepts in figure captions, and make sure that the names in the figures correspond with the names in the captions, as suggested by the reviewer.

9. Minor comments

We will address the Minor Comments of the reviewer by making the suggested changes in the manuscript.

References

S. Chakrabarti, T. Bongiovanni, J. Judge, L. Zotarelli, and C. Bayer. Assimilation of SMOS Soil Moisture for Quantifying Drought Impacts on Crop Yield in Agricultural Regions. IEEE Journal of Selected Topics in Applied Earth Observations and Remote Sensing, 7(9):3867–3879, Sept. 2014. ISSN 2151-1535. doi: 10.1109/JS-TARS.2014.2315999. Conference Name: IEEE Journal of Selected Topics in Applied Earth Observations and Remote Sensing.

Z. Hao and A. AghaKouchak. Multivariate Standardized Drought Index: A parametric multi-index model. Advances in Water Resources, 57:12–18, July2013. ISSN 0309-1708. doi: 10.1016/j.advwatres.2013.03.009.

J. Martínez-Fernández, A. González-Zamora, N. Sánchez, and A. Gumuzzio. A soil water based index as a suitable agricultural drought indicator. Journal of Hydrology, 522:265–273, Mar. 2015.ISSN 0022-1694. doi:10.1016/j.jhydrol.2014.12.051.

J. Martínez-Fernández, A. González-Zamora, N. Sánchez, A. Gumuzzio, and C. M. Herrero-Jiménez. Satellite soil moisture for agricultural drought monitoring: Assessment of the SMOS derived Soil Water Deficit Index. Remote Sensing of Environment, 177:277–286, May 2016. ISSN 0034-4257. doi:10.1016/j.rse.2016.02.064.

V. Sridhar, K. G. Hubbard, J. You, and E. D. Hunt. Development of the Soil Moisture Index to Quantify Agricultural Drought and Its "User Friendliness" in Severity-Area-Duration Assessment. Journal of Hydrometeorology, 9(4):660–676, Aug. 2008. ISSN 1525-755X. doi: 10.1175/2007JHM892.1. Publisher: American Meteorological Society.

[Figure]

**Fig. 1.** Skill scores over grassland pixels, as defined in the supplementary material. An 80% threshold was used.

**Fig. 2.** Skill scores over forested pixels, as defined in the supplementary material. An 80% threshold was used.

---

## Author Comment (AC3) · 19 Feb 2021

We thank the referee for his comments on our paper and welcome this lively discussion of the topic agricultural drought.

The referee addresses several issues, which can be summarized as follows. The referee states that this research manuscript could be specified as a discussion/review rather than a research paper. Secondly, he feels that the pitch of the paper does not hold up well, based on 3 issues with our argument throughout the paper. Finally, there are some explanations of the found differences that are not touched upon in the manuscript, which we will address shortly in this reply.

[Figure]

While we disagree with the reviewer about the novelty of the work (in our view at least part of the analysis, namely the skill score part, is novel, but we agree that many others have looked at satellite soil moisture and NDVI from different perspectives), we see the point that the work might as well be presented as an Opinions paper. Indeed part of our goal with this work was to start a debate, but we simply did not consider the option to submit as HESS Opinions. We hope to hear from the other reviewers how they feel about this suggestion.

The second statement of the referee requires a somewhat longer response, which we will provide here. The three issues the referee gives will be addressed separately, as in the review.

1. We are aware that many different definitions for agricultural drought (AD) exist, as well as many different AD indices. The definition we give in our manuscript ("a soil moisture deficit severe enough to hamper vegetation growth") is based on the knowledge that, although the exact phrasing may differ from author to author, a combination of soil moisture and vegetation is a common denominator. We cited Wilhite and Glantz (1985), who state: "Agricultural drought definitions link various characteristics of meteorological drought to agricultural impacts", and then continue their review citing earlier studies who define ADs slightly differently, though always relating water status to vegetation state. Similarly, Tallaksen & Van Lanen (2004) provide the definition: "The term agricultural drought is used when soil moisture is insufficient to support crops." The references the referee provides treat different AD indices, and can be added to a newer version of the manuscript. We would like to stress that the aim of this manuscript was not to add yet another definition of ADs to the vast literature on this topic, as correctly stated by the referee, but rather to clarify to the public that the current definition of ADs can be confusing and that the term should be avoided. In a possible newer version of the manuscript we will make sure to clarify this.

2. The referee states that from our manuscript, it is unclear to him how the current definition of AD is ambiguous. For the conclusion of our paper to come across well,

it is indeed important that said statement is clear. We are sorry to hear that this is currently clearly not the case. Here, we will try to clarify our statement, and will make sure in a possible revision to do the same.

The referee mentions that by stating that "a soil moisture deficit can lead to increased NDVI" we do not undermine the AD definition, as it is possible that we have been looking at soil moisture deficits that were not enough to hamper vegetation growth. That is indeed possible, and in fact this is in line with the many reports highlighting enhanced vegetation growth and ET in regions that experience drought, but where the soil moisture deficits are not low enough below the critical moisture content to cause a negative impact on vegetation and ET (see Jolly et al., 2005, Teuling et al., 2013, Mastrotheodoros et al., 2020). However, the literature is full of examples where agricultural droughts were studied just by analysing either soil moisture data (see our references in our reply to Reviewer 2) or vegetation greenness. In our manuscript we aim to show that inferring vegetation state from soil moisture levels is not always accurate, for instance when low soil moisture levels lead to enhanced vegetation growth, instead of hampered vegetation growth. This is now more clearly illustrated in a revision of Figure 1 in the manuscript (Fig. 1). Moving away from the definition "agricultural drought" towards "soil moisture drought" and "vegetation drought" would avoid any confusion, both for the public and fellow scientists.

3. The third point the referee makes is based around the lag from low soil moisture levels to hampered vegetation. In our reply to R1, we included an analysis of this lag, if the referee is interested to see such results. Secondly, we indeed claim that the "soil moisture deficit is severe enough to hamper vegetation growth" at the end of summer. However, we feel that the referee might have missed the point that we were trying to make here, namely that this is not always the case, and thus that the definition of AD is ambiguous. This is especially problematic when someone uses the term AD to describe soil moisture droughts.

The final point the reviewer makes is that there are some missed opportunities to explain the differences that we found in our results, and we thank him for pointing us towards relevant literature. We will be sure to touch upon these in a possible revision of our manuscript.

REFERENCES

William M. Jolly et al., 'Divergent Vegetation Growth Responses to the 2003 Heat Wave in the Swiss Alps', Geophysical Research Letters 32, no. 18 (2005), https://doi.org/10.1029/2005GL023252.

Theodoros Mastrotheodoros et al., 'More Green and Less Blue Water in the Alps during Warmer Summers', Nature Climate Change 10, no. 2 (2020): 155–61, https://doi.org/10.1038/s41558-019-0676-5.

Lena M. Tallaksen and Henny A. J. van Lanen, eds., Hydrological Drought – 1st Edition, vol. 84, Developments in Water Science (Elsevier, 2004).

Adriaan J. Teuling et al., 'Evapotranspiration Amplifies European Summer Drought', Geophysical Research Letters 40, no. 10 (2013): 2071–75, https://doi.org/10.1002/grl.50495.

Donald A Wilhite and Michael H Glantz, 'Understanding the Drought Phenomenon: The Role of Definitions', WATER INTERNATIONAL 10, no. 3 (1985): 111–20.

[Figure]

[Figure]

**Fig. 1.** Illustration that different soil moisture climatologies can potentially mean very different things in terms of vegetation.

[Figure]